# Potent Protective Immune Responses to *Senecavirus* Induced by Virus-Like Particle Vaccine in Pigs

**DOI:** 10.3390/vaccines8030532

**Published:** 2020-09-15

**Authors:** Suyu Mu, Shiqi Sun, Hu Dong, Manyuan Bai, Yun Zhang, Zhidong Teng, Mei Ren, Shuanghui Yin, Huichen Guo

**Affiliations:** 1Lanzhou Veterinary Research Institute, Chinese Academy of Agricultural Sciences, Lanzhou 730046, China; mutou6633937@126.com (S.M.); sunshiqi@caas.cn (S.S.); donghu.0608@163.com (H.D.); baimanyuan@caas.cn (M.B.); zhangyun03@caas.cn (Y.Z.); tengzhidong163@163.com (Z.T.); Mei.Ren@student.uliege.be (M.R.); yinshuanghui@caas.cn (S.Y.); 2College of Animal Science, Yangtze University, Jingzhou 434025, China

**Keywords:** Senecavirus A, vaccine, virus-like particles, animal infection model

## Abstract

Senecavirus A (SVA) is the pathogen that has recently caused porcine idiopathic vesicular disease (PIVD). The clinical symptoms of PIVD are similar to those of acute foot-and-mouth disease and also can result in the death of newborn piglets, thus entailing economic losses. Vaccine immunization is the most effective way to prevent and control SVA. Among all SVA vaccines reported, only the SVA inactivated vaccine has been successfully developed. However, to ensure the elimination of this pathogen, safer and more effective vaccines are urgently required. A virus-like particles (VLPs)-based vaccine is probably the best alternative to inactivated vaccine. To develop an SVA VLPs vaccine and evaluate its immune effect, a prokaryotic expression system was used to produce SVA capsid protein and assemble VLPs. The VLPs were characterized by affinity chromatography, sucrose density gradient centrifugation, ZetaSizer and transmission electron microscopy. Meanwhile, the SVA CH-HB-2017 strain was used to infect pigs and to determine infection routes and dose. Experimental pigs were then immunized with the SVA VLPs vaccine emulsified in an ISA 201 adjuvant. The results showed that the VLPs vaccine induced neutralizing and specific antibodies at similar levels as an inactivated SVA vaccine after immunization. The level of INF-γ induced by the VLPs vaccine gradually decreased—similar to that of inactivated vaccine. These results indicated that VLPs vaccine may simultaneously cause both cellular and humoral immune responses. Importantly, after the challenge, the VLPs vaccine provided similar levels of protection as the inactivated SVA vaccine. In this study, we successfully obtained novel SVA VLPs and confirmed their highly immunogenicity, thus providing a superior candidate vaccine for defense and elimination of SVA, compared to the inactivated vaccine.

## 1. Introduction

Senecavirus A (SVA)—also known as *Seneca Valley virus* (SVV)—was isolated from contaminated PER.C_6_ cells in 2002 [1]. This virus was first reported associated with porcine idiopathic vesicular disease (PIVD) in Canada [2]. PCR identification and virus isolation confirmed that several samples from diseased animals contained SVA, but no other pathogens. Later studies found that the clinical symptoms of PIVD caused by SVA that were very similar to those of foot-and-mouth disease (FMD), swine vesicular disease (SVD), swine vesicular herpes and swine vesicular stomatitis virus. Because FMD is categorized by the World Organization for Animal Health (OIE) as a highly infectious acute disease requiring an immediate report, SVA also has attracted great attention. Thus, far, SVA has spread widely in Canada [2], USA [3], Brazil [4], China [5], Thailand [6], Colombia [7] and Vietnam [8]. According to an epidemiological study, the morbidity rate of piglets infected with SVA is about 2–3 times higher than that of unaffected piglets, posing a great challenge for pig production [9]. SVA inhibits IFN-γ production in the host via protease 3C^pro^, thus destroying the host’s defense system [10]. Therefore, the prevention and control of this virus are urgently required.

Vaccination is still the most effective way to control and prevent infectious diseases. Currently, most available vaccines are inactivated or live-attenuated pathogens [11]. Only an inactivated vaccine has been developed for SVA so far. The inactivated vaccine induced high neutralizing antibody titers, thus providing specific protection against SVA [12]. Although inactivated vaccines are very useful, insufficient inactivation of virus during the vaccine production may lead to the spread of disease. Therefore, the development of a new type of vaccine with high safety is urgently required [13].

Many types of genetically engineered vaccines have been produced, including deletion vaccines, peptide vaccines, gene vaccines and virus-like particles (VLPs). VLPs are empty viral particles composed of viral structural proteins with a self-assembly capacity. These proteins spontaneously form a three-dimensional structure resembling the virus, which exposes multiple antigenic sites of the virus and can be taken up by dendritic cell, thus triggering an effective immune response [14]. Therefore, VLPs have many advantages of among these vaccines mentioned above. VLPs are safer than live viruses and gene-deleted vaccines because they do not contain genetic material and are not infectious or replication-competent [15]. Unlike peptide vaccines, VLPs exert the same immunogenicity as the intact virus and induce perfect immune protection [13]. To date, VLPs-based vaccines have been developed as approved vaccine products for *Human papillomavirus* [16] and *Hepatitis B virus* [17,18]. Thus, VLPs are also expected to be the ideal vaccine for SVA [19].

*Escherichia coli* (*E. coli*) has been widely used as an expression system for VLPs [17,20,21,22,23] and is quite popular for the large-scale expression of proteins because it is highly scalable at low cost [24]. In previous studies, we have successfully developed VLPs of *Foot-and-mouth disease virus* (FMD), *Porcine circovirus 2* (PCV2), *Rabbit hemorrhagic fever virus* (RHDV), *Porcine parvovirus* (PPV) and *Canine parvovirus* (CPV) using an *E. coli* expression vector [20,21,22,23]. Since FMDV and SVA are both members of the family of small RNA viruses, our previous studies would be useful for the construction of SVA VLPs [21].

SVA is a positive-sense single-stranded RNA virus belonging to the genus *Senecavirus* in the family Picornaviridae. It has the typical L-4-3-4 structure of the small RNA viruses and a diameter of about 25–30 nm [25]. A single polyprotein is encoded by 7200-nucleotides-long positive-sense single-stranded RNA, then the polyprotein is cleaved to produce the P1 region containing the structural proteins and the P2 and P3 regions containing the nonstructural proteins (2A, 2B, 2C, 3A, 3B, 3C and 3D) [26]. The P1 region is cleaved to form VP0 (43 kDa), VP3 (31 kDa) and VP1 (27 kDa) of the viral capsid. After the virus matures, VP0 gradually generates VP2 (35 kDa) and VP4 (7 kDa) [25,27] and VP2 contains the main antigenic site of SVA [28].

Based on our previous studies [20,21], we used an improved SUMO fusion protein prokaryotic expression system to generate His–SUMO fusion proteins, which are recognized by a small ubiquitination enzyme. The native conformation of VLPs was achieved by removing the tags from the fusion protein with a small ubiquitination enzyme [29,30]. This allowed the target protein to spontaneously form SVA VLPs in vitro that were similar in size and shape to the native SVA particles. SVA VLPs is emulsified by using adjuvant ISA 201, inject intramuscularly with one dose. Both humoral (specific antibodies and neutralizing antibodies) and cellular (IFN-γ) immune responses were activated, which were sufficient to protect the host pigs from SVA infection. In this study, we undertook to develop a safer and more effective candidate SVA vaccine and to provide a way to eliminate SVA without using the attenuated and inactivated vaccine.

## 2. Results

### 2.1. Infectious Routes and Doses of SVA CH-HB-2017 Strain

By observing the incidence of pigs with three different routes of infection, as shown in Table 1, it is found that all pigs infected SVA via intramuscular (3 mL, 7 × 10^7.8^ PFU/mL) and intranasal (1.5 mL each nostril, 7 × 10^7.8^ PFU/mL) presented clinical signs (Figure 1A). The clinical scores of each pigs were recorded, and the viral RNA was detected in the blood (pigs 11–15) with qRT–PCR. As shown in Figure 1A, the highest level of virus RNA was detected at 3 dpi, Afterwards, it gradually declined and was undetectable at 7 dpi. Vesicles appear on the nose or hoof at 4 to 6 dpi, then ulcerated at 7 dpi (Figure 1B). The mean rectal temperature was higher than that of the control group (Figure 1C).

### 2.2. Expression and Purification of SVA Capsid Proteins

To express the SVA structural capsid proteins VP0, VP1 and VP3, recombinant plasmids encoding VP0 (pSMAVP0), VP1 (pSMKVP1) and VP3 (pSMKVP3) were constructed, transformed into the expression bacteria BL21 (DE3) and induced expression. Fusion proteins were purified with a previously described method [20]. The molecular weights of the fusion proteins His6-Sm-VP0, His6-Sm-VP1 and His6-Sm-VP3 were about 51, 41 and 39 kDa, respectively (Figure 2A, Lanes 1,3).

The His–SUMO fusion tag was cleavage with the SUMO protease and final cleavage products included a His-tagged band at 12 kDa (His6-Sm tag) and three water-soluble polypeptides with molecular weights of 39, 29 and 27 kDa, respectively (Figure 2A, Lanes 2,4).

### 2.3. Characterization and Quantification of SVA VLPs

To test whether recombinant VP0, VP1 and VP1 could assemble into VLPs, the samples were separated by density gradient centrifugation. The optical density at wavelength of 280 nm (OD_280_) showed two distinct peaks curve (Figure 2B), with the hydrated diameters 12.19 nm or 33 nm, respectively (Appendix A). Transmission electron microscopy showed that the diameters of particles obtained from two peak fraction were 13 nm or 30 nm (Figure 2B). This result indicated that the fusion proteins successfully assemble into VLPs, but also assemble into some intermediates (pentamers) [25]. The peak fractions were pooled and the percentage of VLP was calculated from the peak area. This portion was approximately 39.8% of the total expressed proteins (Figure 2B), while the other 60.2% of proteins failed to assemble into stable empty capsids (Figure 2B).

### 2.4. Immunogenic Evaluation of SVA VLPs Vaccine in Pigs

Serum specific antibody levels were detected by indirect-ELISAs at 14, 21, 35 and 49 days after immunization. As shown in Figure 3A, the similar SVA-specific antibody levels were obtained in VLPs group and the inactivated group. The antibody levels of group vaccinated with inactivated virus slightly decreased at 49 days compared to VLPs group (*p* < 0.01). In the micro neutralization assay, it showed that higher neutralizing antibodies induced both by inactivated and VLPs vaccine (Figure 3B).

The degrees of cellular immune responses (Th1) and humoral immune responses (Th2) are represented by the levels of IgG2a and IgG1 antibodies, respectively. Serum IgG1 and IgG2a antibodies levels were detected by ELISAs at 14, 21, 35 and 49 days after immunization (Figure 3C,D). The IgG1 antibody levels obtained by the inactivated vaccine was significantly higher than that of the VLPs vaccine at 14 (*p* < 0.01), 21 (*p* < 0.01), 35 (*p* < 0.05) and 49 (*p* < 0.01) days after immunization. However, The IgG2a antibody level of group immunized with VLPs vaccine were similar to inactivated vaccine at 14 and 21 days after immunization (*p* = n.s.). However, the antibody levels of group immunized with inactivated vaccine slightly increased at 35 (*p* < 0.05) and 49 days (*p* < 0.0001) compared to VLPs vaccine group. This result indicated that strong humoral immune responses were induced by the VLPs vaccine and inactivated vaccine and more significant in the inactivated vaccine.

### 2.5. SVA VLPs Vaccine-Induced INF-γ Response

To further evaluate cellular immune response induced by VLPs vaccine, the IFN-γ levels were detected by ELISA. As shown in Figure 4, there is no difference between groups immunized with inactivated vaccine and VLPs vaccine for IFN-γ titer at 14, 21, 35 days (*p* = ns). However, the duration of cellular immunity induced by inactivated vaccine is significantly longer than that of the VLPs vaccine (*p* < 0.01).

### 2.6. Protection against SVA Challenge

To determine the protective effect of the VLPs vaccine in pigs, all pigs were challenged with SVA CH-HB-2017 strain via intramuscular (3 mL, 7 × 10^7.8^ PFU/mL) and intranasal (1.5 mL into each nostril, 7 × 10^7.8^ PFU/mL) routes at 49 days after immunization. Clinical signal was recorded and scored according to Table 1. As shown in Table 2, five pigs injected with PBS appeared clinical symptoms within three days after challenge. Lesions were observed on foot hooves of all pigs. Pigs vaccinated with VLPs and the inactivated vaccine has no clinical signs and no RNA in blood samples was detected.

## 3. Discussion

SVA mainly causes blisters on the noses and hoofs of pigs, so it is challenging to distinguish it from diseases such as FMD, porcine vesicular disease and vesicular stomatitis virus. VLPs vaccines have very highly safe and immunogenic and are considered as the best option for elimination, prevention and control of virus infection. In this study, it showed that the vector-encoding SUMO protein successfully expressed three structural proteins of SVA in *E. coli*. After the His–SUMO tag was removed, we confirmed the successful assembly of the protein complex by characterizing the VLPs with sucrose density gradient centrifugation, transmission electron microscopy and nanoparticle sizing. We demonstrated that the VLPs produced in *E. coli* were similar in morphology to the empty particles in SVA-infected cells. The large-scale expression of soluble proteins is required to produce SVA VLPs. The SUMO enhances the expression of fusion target protein, improves its folding. It can be efficiently cleaved by a protease, so it is preferred for the production of difficult-to-express proteins [31,32], including those of PCV2 [20], RHDV [22] and FMDV [21]. Although the process mechanism of sumo protein in *E. coli* is not yet clear, the expression of soluble proteins may be related to the hydrophilic surface and hydrophobic core of SUMO, which have a similar effect on insoluble proteins as detergent [31]. The production strategy for VLPs using *E. coli* has also been used in the production of FMDV VLPs and both viruses belong to the small RNA virus family. This strategy allowed the output of FMDV capsid of similar size and structure to the native virus capsid [23]. Moreover, the combination of FMDV VLPs and adjuvant 206 generated similar antibodies and immunological protection as the inactivated vaccine in guinea pig, pigs and cows [21]. These results are consistent with the rate of protection generated by the SVA VLPs in the present study.

The SVA VLPs generated neutralizing and specific antibodies at similar levels in the pigs, as the inactivated vaccine which was used as the positive control. This is an essential factor in the elimination of viral infection by B lymphocytes. The IgG1/IgG2a ratio also demonstrated that this vaccine prone to induce the B lymphocytes response, whereas the induction of IFN-γ implies a cellular immune response. Both humoral and cellular immunity are necessary for the prevention of viral infections [15]. A viral challenge test could confirm the effectiveness of the vaccine. Therefore, the SVA VLPs vaccine was considered effective in protecting pigs from SVA infection. In this study, the VLPs-immunized pigs produced very high levels of both specific and neutralizing antibodies, which are the main factor of humoral immunity. The high antibody levels were consistent with previously reported antibody levels after SVA infection [28]. However, FMDV, which also causes vesicular disease, does not induce such high antibody levels [21]. This may be because the major antigenic site of SVA are in VP2, whereas those of FMDV is in VP1 [28]. The Th1-type cellular immune response caused by IFN-γ is mainly mediated by T lymphocytes, which primarily destroy endogenous antigens and are essential for viral clearance. In the present study, the VLPs induced similar levels of INF-γ than the inactivated vaccine (*p* = ns) before 35 days after immunization. This virus mainly infects the host by inhibiting the production of INF-γ to escape the immune response [10]. It mainly exists the tonsils of pigs after infection [33], leading to continuous infection of the host. Therefore, increasing the immune response of Th-1 type cells is key to improving the vaccine protection rate. In this study, the inactivated vaccine (10 μg) and the VLPs vaccine (50 μg) can full protected pigs from SVA infection. However, a more suitable immunization dose was not determined, future studies should focus on the development of excellent adjuvants and suitable immunization dose, as well as the optimization of the infectious dose.

The pathogenic mechanism of SVA has been reported [34], but the dose and routes of viral infection have not yet been established. However, the development of vaccines requires both a suitable challenge dose and infection route. Different strains may also favor different paths of infection. For example, oronasal infection has been commonly used in previous studies. A dose of 5 mL of a virus suspension containing 7 × 10^−7.07^ PFU/mL via the oral (2 mL) and intranasal (1.5 mL to each nostril) at a can result in successful SVA infection, although this is a very high dose [34]. However, in our study, infection via oronasal was failed at this dose (not published), which may be attributable to differences in the procedures and strains used. The animal experiments is very costly, hence, to avoid of failure, we changed the infection strategy and used nasal (1.5 mL into each nostril, 7 × 10^7.8^ PFU/mL) and intramuscular (3 mL, 7 × 10^7.8^ PFU/mL) injections to achieve 100% infection. An intramuscular injection of 6 mL (7 × 10^7.8^ PFU/mL) alone was not ideal because it only reached 80% infection. There is no report about SVA infection via the intramuscular routes, which may be related to its slow absorption and the complex bio environment. SVA infection via the marginal ear vein is theoretically the fastest way to infect pigs, but the complexity of the operation makes it very challenging when more than 3 mL (7 × 10^7.8^ PFU/mL) dose used, although it surprisingly achieved an infection rate of 80%. Our results suggested that it may be future direction to further concentrate viral sample in order to achieve SVA infection in pigs and developed typical symptom via marginal ear vein.

## 4. Materials and Methods

### 4.1. Viruses and Cells

SVA strain CH-HB-2017 (The number of plaques forming units: 7 × 10^7.8^ PFU/mL) (GenBank accession number: MN922286) was previously isolated by our laboratory [35]. IBRS-2 cells were cultured in Dulbecco’s modified Eagle’s medium (DMEM; Gibco, Waltham, MA, USA) supplemented with 10% fetal bovine serum (FBS; HyClone, Mordialloc, Australia), 100 U/mL penicillin and 100 mg/mL streptomycin, at 37 °C in a 5% CO_2_ atmosphere.

### 4.2. Preparation and Characterization of SVA VLPs

Capsid proteins VP0, VP1 and VP3 of SVA were expressed in *E. coli*, as reported previously [20,21,22]. The recombinant proteins were isolated and purified with a His-tag affinity column and then dialyzed against buffer (500 mM NaCl, 40 mM Tris-HCl, 2 mM MgCl_2_, 10% glycerin, pH 8.0) with SUMO protease for 36 h at 4 °C. The target proteins were analyzed by SDS-PAGE and western blot (Antibody with a porcine polyclonal antibody).

The samples were loaded onto the top of a 10–50% sucrose density gradient prepared with Gradient Station™ (Bio Comp Instruments, Inc., Fredericton, NB, Canada) in a 13.2 mL open-top tube and ultracentrifuge (XPN-100, Beckman) at 110,000× *g* for 3 h at 4 °C in an SW41 Ti rotor. Twenty-two fractions were separated and analyzed with a BioMate™ 3S UV-vis spectrophotometer (Thermo Fisher Scientific, Inc., Waltham, MA, USA). After fitting the peaks for the pentamer and VLPs as two gaussians, the two areas were calculated separately and take the area of the VLPs over the sum. Furthermore, the peak fractions and original samples were detected by the BCA protein assay kit. The peak fractions were further analyzed using a Zetasizer Nano (Malvern Panalytical Ltd, Grovewood Road, UK) equipped with a photodiode laser (830 nm). The dispersant refractive index and viscosity were assumed to be 1.33 and 1.003 P, respectively [21]. To observe the particle morphology, the peak fraction sample was adsorbed onto 300 mesh formvar-coated copper grids (Polysciences, Inc., Warrington, PA, USA) and stained with 2% phosphotungstic acid (pH 7.6) for 1 min. The grids were examined by transmission electron microscopy (H-7100FA, Hitachi) at an acceleration voltage of 100 kV.

### 4.3. Vaccine Preparation

The SVA inactivated vaccine was prepared according to previous studies [36,37]. In brief, the virus was collected and extracted, after IBRS-2 cells were infected with SVA for 10 h (until 100% cytopathic effect). The viral suspension was inactivated by binary ethylenimine at 30 °C for 28 h. The inactivation effect of the virus is verified by reverse transcription (RT)-quantitative PCR (qPCR). The Inactivated virus was further purified by ultracentrifugation and major viruses peak was quantitated by the BCA protein assay kit.

VLPs and inactivated virus were emulsified with equal volumes of ISA 201 adjuvant (Montanide). ISA 201 was used as the oil phase. The oil phase and inactivated SVA fluid or VLPs were homogenized with a T 10 basic Ultra-Turrax^®^ homogenizer (IKA, Beijing, China) at 8000 rounds per minute for 30 min.

### 4.4. Ethics Statement

All animals received humane care in compliance with good animal practices, according to the Animal Ethics Procedures and Guidelines of the People’s Republic of China. The specific experiments were approved by the Animal Ethics Committee of Lanzhou Veterinary Research Institute, Chinese Academy of Agricultural Sciences (permit number LVRIEC2018-008).

### 4.5. Infection and Immunization of Pigs

To establish the optimal dose and routes of SVA infection in pigs, 15-week-old finishing pigs (approximately 50 kg each) were purchased from conventional breeding/fattening farms and kept in five rooms, with access to food and water ad libitum. They were negative for both FMDV and SVA were antibodies at the start of the experiment. The pigs ( total 23 pigs ) were randomly assigned into five groups: Intramuscularly (3 mL, 7 × 10^7.8^ PFU/mL ) and intranasal (1.5 mL into each nostril, 7 × 10^7.8^ PFU/mL) group (*n* = 5) were infected with SVA CH-HB-2017 strain; Intramuscularly (1.5 mL, 7 × 10^7.8^ PFU/mL ) and intranasal (0.75 mL into each nostril, 7 × 10^7.8^ PFU/mL) group (*n* = 5) were infected with SVA CH-HB-2017 strain; Intramuscularly group (*n* = 5) were infected with SVA CH-HB-2017 strain (6 mL, 7 × 10^7.8^ PFU/mL); Ear vein group (*n* = 5) were infected with SVA CH-HB-2017 strain (3 mL, 7 × 10^7.8^ PFU/mL); and negative control (*n* = 3) were infected with DMEM. All clinical symptom and rectal temperature were recorded daily from 0 to 10 days post-infection (dpi), the scores of clinical symptoms are shown in Table 3. The blood samples were collected at 1, 3, 5, 7 and 9 dpi.

To evaluate the efficiency of the SVA VLPs vaccine, 9-week-old finishing pigs (approximately 35 kg each) were purchased from conventional breeding/fattening farms and kept in three rooms, with access to food and water ad libitum. They were all negative for FMDV and SVA antibodies at the start of the experiment. The 15 pigs were randomly assigned to three groups: inactivated vaccine group (*n* = 5) was injected intramuscularly with 2 mL of inactivated SVA vaccine (10 μg). The VLPs vaccine group (*n* = 5) was injected intramuscularly with 2 mL of SVA VLP (50 μg) vaccine. Pigs in control group (*n* = 5) were injected with PBS. All pigs were boost vaccinated at 21 days. Animal serum samples were collected at 0, 14, 21, 35 and 49 days and stored at −80 °C. At 49 days, all pigs were challenged with SVA CH-HB-2017 strain via the intramuscular (3 mL, 7 × 10^7.8^ PFU/mL) and intranasal routes (1.5 mL into each nostril, 7 × 10^7.8^ PFU/mL).

### 4.6. Detection of Specific Antibodies, IgG1 and IgG2a

Specific antibodies, IgG1 and IgG2a, were detected in each serum sample with AsurDx™ Senecavirus A (SVA) Antibody Test Kit (Biostone™ Animal Health, Dallas, TX, USA), Porcine Immunoglobulin G1 (IgG1) ELISA Kit (OmnimAbs, Alhambra, CA, USA) and Porcine Immunoglobulin G2a (IgG2a) ELISA Kit (OmnimAbs, Alhambra, CA, USA), respectively, according to the manufacturer’s instructions.

### 4.7. Neutralization Assay

Serum samples were inactivated at 56 °C for 30 min. Serum neutralization assays were then performed on IBRS-2 cells. Diluted serum samples were prepared starting with 1:4, repeating 4 times for each dilution. The diluted sera were incubated with 100TCID50 of SVA strain CH-HB-2017 for 1 h at 37 °C. IBRS-2 cells (10^5^ cells/mL, 50 μL) were then added to the wells of 96-well plates, cultured in an incubator at 37 °C under 5% CO2 for 72 h and observed for cytopathic effects (CPEs) under an inverted microscope. The neutralizing antibody titer was calculated by the Reed–Muench method. In brief, 100% CPE of cells in the tested serum wells is judged as negative, and more than 50% of cells are protected as positive. Finally, calculate the serum dilution that can protect 50% of the cell wells from cytopathic, and this dilution is the serum neutralizing antibody titer.

### 4.8. Detection of Cytokines Interferon-γ

Cytokines were detected in each serum sample with porcine interferon-γ (IFN-γ) ELISA kit (OmnimAbs, Alhambra, CA, USA) according to the manufacturer’s instructions. In brief, samples that included standards of known concentrations and unknowns were pipetted into coated microtiter wells. After incubating, biotinylated anti-IgG and combined streptavidin-HRP were added. After washing completely, TMB substrate solution, TMB chromogen solution was added until it became blue when the HRP enzyme-catalyzed. The optical density (OD) was measured at 450 nm with microtiter plate reader, calculate IFN-γ concentration by standard curve.

### 4.9. qRT-PCR

Total RNA was extracted from the anticoagulated blood samples with the TaKaRa RNAiso Plus Extraction Kit (Takara Bio, Inc., Kusatsu, Japan), according to the manufacturer’s instructions. SVA RNA was detected with qRT–PCR targeting the SVA L/VP4 polymerase gene, and the limit of detection was 64 copies/μL, as previously described [38]. Briefly, reaction was set in a 25-μL volume containing 12.5 μL of 2 × TB Green Premix Ex Taq II (Tli RNaseH Plus) Master Mix (Takara Bio, Inc.), 2 μL of cDNA or DNA template, 8.5 μL of diethylpyrocarbonate (DEPC)-treated water and 1 μL (the stock concentration 10 μM) of each primer (SVA-FP1 (5′-TGAAGTTGCGGAGAAGAT-3′) (location: 878–896) and SVA-RP1 (5′-TTGCGTAGTAATTGAAGGT-3′) (location: 966–985) ( GenBank accession no. MN922286)). Amplification conditions included an initial denaturation step at 95 °C for 30 s, followed by 40 cycles at 95 °C for 15 s and 60 °C for 30 s. Amplification was performed using the CFX96 Touch™ real-time PCR detection system (Bio-Rad Laboratories, Hercules, CA, USA). Melting curve analysis was performed using Bio-Rad CFX96 software.

### 4.10. Statistical Analysis

All statistical analyses were conducted with a 2-way ANOVA in the GraphPad Prism 8.0.1 (GraphPad Software, San Diego, CA, USA). Ns were considered no statistically significant. Differences with 0.01 < *p* ≤ 0.05 were considered statistically significant and those with *p* ≤ 0.01 were considered highly significant (* *p* < 0.05, ** *p* < 0.01, *** *p* < 0.001, **** *p* < 0.0001). Gaussian analyses were used OriginPro 8.5.1 (OriginLab Corporation, Northampton, MA, USA) to calculate the assembly ratio of VLPs and pentamers.

## 5. Conclusions

In summary, we produced SVA VLPs with a low cost, safe and widely applicable strategy. Immunological study showed that the VLPs vaccine induced both humoral and cellular immune responses in pigs, which then resisted SVA infection. These results suggested that SVA VLPs constituted a desirable candidate vaccine against SVA infection.

## Figures and Tables

**Figure 1 vaccines-08-00532-f001:**
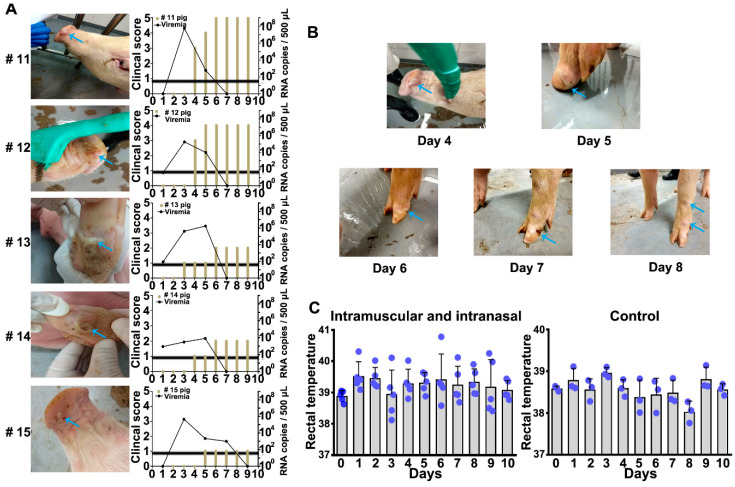
Clinical symptoms and rectal temperature after Senecavirus A (SVA) infection in pigs. (**A**) Pigs infected intramuscularly (3 mL, 7 × 10^7.8^ PFU/mL) and intranasally (1.5 mL into each nostril, 7 × 10^7.8^ PFU/mL). Clinical symptoms of pigs were recorded during the test and scored in order. Pigs 11,12, and 15 had blisters in the nose; 13 and 14 had blisters in the hoof. Serum was collected and the virus content was detected using qRT-PCR technology; (**B**) clinical symptoms after SVA CH-2017-HB strain infection. Pigs developed blisters in the nose at four days and white blisters in the hoof at five and six days. Blisters ulcerated at seven days and the joints were red and swollen. The rupture of the hoof gradually returned at eight days; (**C**) rectal temperature 1–10 days after SVA infection in pigs (Intramuscularly (3 mL, 7 × 10^7.8^ PFU/mL ) and intranasal (1.5 mL into each nostril, 7 × 10^7.8^ PFU/mL) group (*n* = 5)). All clinical changes indicated by blue arrows. Black line represents the cutoff value of the qRT-PCR technology detection.

**Figure 2 vaccines-08-00532-f002:**
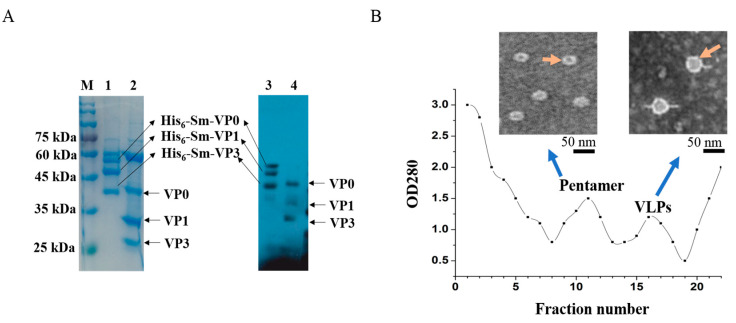
Expression and assembly of SVA capsid protein. (**A**) SDS-PAGE (Lanes 1, 2) and western blot (Lanes 3, 4). Lanes 1, 3: SVA purified capsid-protein complexes His_6_-sm-VP0, His_6_-sm-VP1 and His_6_-sm-VP3. Lanes 2, 4: Target proteins VP0, VP1, VP3 purified after protease digestion; (**B**) SVA VLPs purified by sucrose gradient ultracentrifugation and the OD_280_ fluorescence absorption measured by spectrophotometer. Negative-staining electron micrographs of SVA pentamer (right, orange arrow) and VLPs (left, orange arrow). Scale bar = 50 nm.

**Figure 3 vaccines-08-00532-f003:**
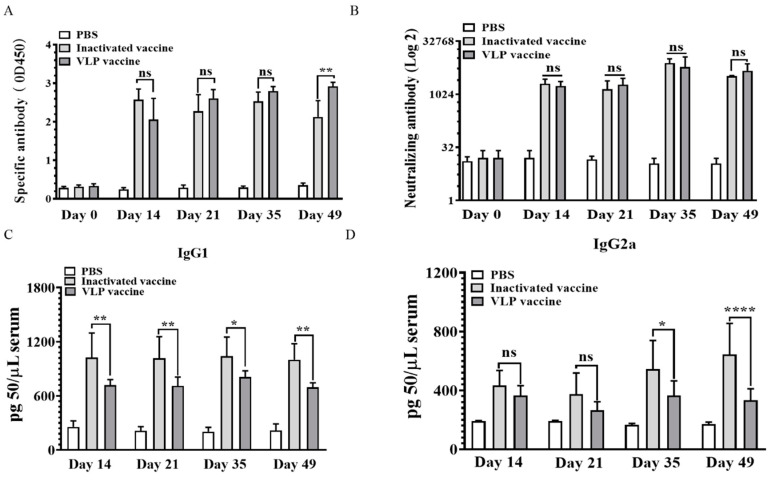
Detection of serum antibodies. (**A**) Specific antibodies; (**B**) neutralizing antibodies; (**C**) IgG1 antibodies; (*D*) IgG2a antibodies. Levels of serum antibody detected in pigs vaccinated with SVA VLPs (*n* = 5), inactivated vaccines (*n* = 5) and PBS (*n* = 5) at 0 days, 14 days, 21 days, 35 days and 49 days post-immunization (ns were considered no statistically significant, * *p* <0.05, ** *p* <0.01, **** *p* <0.0001).

**Figure 4 vaccines-08-00532-f004:**
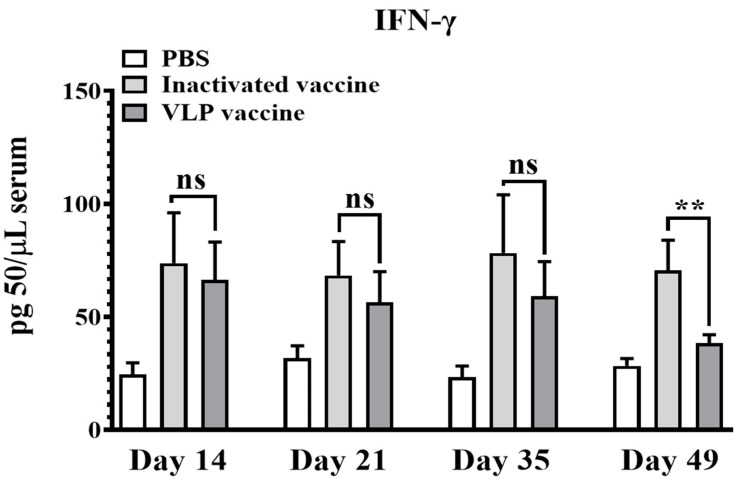
Detection of serum INF-γ. Serum collected from pigs vaccinated with SVA VLPs (*n* = 5), inactivated vaccines (*n* = 5) and PBS (*n* = 5) at 0 days, 14 days, 21 days, 35 days and 49 days post-immunization. INF-γ titers represent the intensity of cellular immune response (ns were considered no statistically significant, ** *p* < 0.01).

**Table 1 vaccines-08-00532-t001:** SVA CH-HB-2017-strain-infected minimum dose exploration.

Inoculation Method (Group)	Pig Number	Inoculation Dose (7 × 10^−7.8^ PFU/mL)	Infected (+/−)	Score	Incidence Rate (%)
**Intramuscular and** **i** **ntranasal**	11	3 ^a^ mL+3 ^b^ mL	+	5	100
	12	3 ^a^ mL+3 ^b^ mL	+	4	
	13	3 ^a^ mL+3 ^b^ mL	+	2	
	14	3 ^a^ mL+3 ^b^ mL	+	1	
	15	3 ^a^ mL+3 ^b^ mL	+	2	
**Intramuscular and** **i** **ntranasal**	16	1.5 ^a^ mL+1.5 ^b^ mL	+	3	60
	17	1.5 ^a^ mL+1.5 ^b^ mL	+	2	
	18	1.5 ^a^ mL+1.5 ^b^ mL	+	3	
	19	1.5 ^a^ mL+1.5 ^b^ mL	−	0	
	20	1.5 ^a^ mL+1.5 ^b^ mL	−	0	
**Intramuscular**	21	6 mL	+	2	80
	22	6 mL	+	3	
	23	6 mL	+	2	
	24	6 mL	+	1	
	25	6 mL	−	0	
**Ear vein**	31	3 mL	+	2	80
	32	3 mL	+	1	
	33	3 mL	+	1	
	34	3 mL	+	1	
	35	3 mL	−	0	
**Control**	50	None	−	0	0
	51	None	−	0	
	52	None	−	0	

^a^: intramuscular. ^b^—nasal. +—positive result for SVA-infected; −—negative result for SVA-infected.

**Table 2 vaccines-08-00532-t002:** Protection against SVA challenge.

Group	Pig Number	Infected (+/−)	Score	Viremia	Level of Protection
Inactivated vaccine	1	−	0	No	100%
	2	−	0	No	
	3	−	0	No	
	4	−	0	No	
	5	−	0	No	
VLPs vaccine	6	−	0	No	100%
	7	−	0	No	
	8	−	0	No	
	9	−	0	No	
	10	−	0	No	
Control	11	+	3	Yes	0
	12	+	2	Yes	
	13	+	2	Yes	
	14	+	2	Yes	
	15	+	2	Yes	

+—positive result for SVA-infected; −—negative result for SVA-infected.

**Table 3 vaccines-08-00532-t003:** Clinical symptom scores for SVA-infected pigs.

Clinical Symptoms	Clinical Score	Total Score
Left forefoot blisters	1	5
Right forefoot blisters	1	
Left hindfoot blisters	1	
Right hindfoot blisters	1	
Mouth and nose blisters	1	

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
