# Peer review of "Potent Protective Immune Responses to Senecavirus Induced by Virus-Like Particle Vaccine in Pigs"

_vaccines, 2020, doi:10.3390/vaccines8030532_

Round 1

Reviewer 1 Report

Mayor observations
1. Please use a translation service as the quality of the English is very poor. All comments here are related to the highlighted text in the attached pdf file.
2. Lines 100 to 101 - It would be good to very briefly describe how the immunization was performed… intravenously, one dosis, adjuvants?? Just a brief description
3. Line 108 - How can you talk about mimics natural infection using vaccination. This makes no sense, perhaps something got lost in the translation. This sentence makes no sense
4. Line 111 - This line makes no sense, you are talking about a vaccine and at the same time you are talking about infection. Do you mean symptoms? It is very confusing. Also, what are groups 1, 2, 3 and 4? These groups need to be defined.
5. Line 112 - The volume of the infection is not important, what it is important is the number of viral particles. You can inoculate with 1L of particles and if they are non-infections or if there is 1 particles per liter nothing will happen. Please clarify this. How did you determine that the infection rate is 100%? This applies to the rest of the paragraph. Talking about volumes is completely irrelevant, the important value is the number of plaque forming units, the multiplicity of infection or the number of genome copies per mL. Those experiments should be in terms of these quantities, otherwise it is impossible to replicate the experiments or to understand what is going on. Furthermore, this shows the the authors have very little understanding about basic virology, which is worrisome. A virologist will never make this mistake.
6. Line 117 - Now the mention group 5 which has been not defined. Please see comment 3.
7. Lines 124 to 125 - How would you observe clinical symptoms from anticoagulantes blood samples?
8. Figure 1 - The resolution of this figure is poor, I cannot see the scale in figure 1D. Also, please draw a line indicating the limit of detection of you assay in figure 1D. That is, the copy number you get with the non-template control.
9. Figures - The resolution of all figures is really bad.
10. Figure 2A and B - While cutting and pasting gels can be an acceptable procedure. What was done in figure 2A and 2B is unacceptable; you cannot copy a lane from a different experiment and then compare it with the ladder of other gel. I cannot accept these two gels; they need to be repeated so there is a ladder in the western blot.
11. Figure 2C - There axis are not labeled. Please label them.
12. Figure 2 - the letters in the figure legend dos not correspond to the images.
13. Line 145 to 146 - Do you mean a z-sizer? What is the polydispersity index of the measurements and standard deviation?
14. Line 147 - What is the standard deviation of the measurements?
15. Section 2.3 - Do the particles contain RNA?
16. Section 2.3 and 2.4 - In previous sections it was mentioned that there were five groups, suddenly this sections only mention three groups. What is going on? Please clarify
17. Lines 153 to 156 - What dosis of vaccine was used? How were the animals immunized (route)?
18. Lines 160 to 161 - How much higher?
19. Lines 162 to 166 - Please give a brief description of the microneutralizaiton assay. More importantly, you needs to specify the dilution of the serums to know how powerful the neutralization activity of the vaccine is. This assay should mention the amount of virus used for the neutralization assay and all the tested dilutions; this is critical information, without it it might be that you need to have a 1:100 dilution of the serum to neutralize the virus for one group and a 1:1000 for the other group. Or it might be that only undiluted serum gives protection, all this information is key to understand to magnitude of the protective response.
20. Lines 172 to 174 - by how much?
21. Lines 183 to 184 - by how much?
22. Lines 187 to 188 - Before the groups were, vaguely, identified by numbers now is by letters. This is totally confusing.
23. Important observation: from the purified sample what is the proportion of assembled VLPs and partially assembled VLPs. This information is key. Also please, explain how you determined this.
24. Besides all the problems mentioned in the previous line and in the following section the authors do not clarify the composition of the vaccine (i.e., VLP concentration) and purity (fraction of proteins as VLPs and as incomplete VLPs in the vaccine). This is a crucial point for the formulation of the vaccine. At least they should mention the total amount of protein in each immunization and how this compares with the inactivated vaccine. Did they use the same amount of protein in the VLPs and inactivated vaccine? If not, why?
25. The lack of experimental information makes impossible to future replicate these results in other labs. Please be careful detailing all experimental information.
26. Lines 231 to 232 - I don’t understand how they can conclude this. These are different viruses.
27. Lines 235 to 236 - this was not clear int he previous sections. Please explicitly say by how much.
28. Line 237 - Virus infect tissues, they do not parasite.
29. Lines 240 to 241 - What does full dose mean? What is the composition of a dose?
30. Lines 250 to 251 - this was not mentioned in the results and should be.
31. Line 254 - The dose of the virus should be expressed in PFU/mL, please convert the TCID 50/ml to PFU/mL.
32. Line 259 to 260 - This statement makes no sense. In the previous sentences they explain that they cannot achieve 100% infection, thus it makes no sense to suggest further dilutions. What you need are more concentrated viral samples.
33. A big pitfall in this article is the lack of information about the ideal vaccine dilution that can protect the pigs.
34. Lines 297 - If the virus was inactivated then how can you make three passages? This data is key, please put in a figure.
35. Lines 300 to 303 - Please see comments 17, 24 and 29.
36. Section 4.5 - The groups are not explain in the main text.,please do it.
37. Lines 316 to 321 - This is fundamental error in the experimental design. They should have used innocuous with different amount of SVA so that each infection protocol delivered the same amount virus. This explain why they were not Abe to have 100% infection using different routes of infection.
38. Lines 328 to 331 - Is the does of inactivated SVA vaccine the same as the VLPs? In other words, are 10^-8.8 the same as 50 ug? If not then this a problem. Please explain how you determined this.
39. Lines 349 to 357 - Please describe the methods.
40. Section 4.9 - Please indicate the limit of detection of this assay.

Editorial comments
1. Line 18 - see comment
2. Lines 22 to 25 - This sentence is hard to understand. The syntaxes should be improved.
3. Line 26 - remove the comma
4. Line 27 - were instead of was (VLPs is plural)
5. Line 28 - What technique do you mean by “nanoparticle detection”?
6. Lines 29 to 30 - I don’t think that parameters of infection pathway is the proper term. Please use a better term to describe what you did. This term is so confusing that it is not clear what do you mean.
7. Line 30 - The proper term is immunized not injected.
8. Line 31 - Is oil the vaccine carrier or the adjuvant? I would think is the vaccine carrier.
9. Line 31 to 32 - Remove the hyphen between vaccine and induced, otherwise the rest of the sentence makes no sense. Perhaps use vaccination or immunization rather than vaccine.
10. Line 35 to 36 - The statement about IgG1 and IgG2 does not belong to an abstract. The important information is contained before the comma.
11. Line 37 - I am not sure that protection rate is the proper term. A rate is the change of parameter with respect of another parameter. Perhaps, levels of protection would be a better term.
12. Line 38 - “Good” immunogenicity is not the right term. What about “is highly immunogenic”?
13. Line 39 - This sentence is incomplete. “Of SVA, compared to the inactivated vaccine”.
14. Line 43 - There should be a reference at the end of the first sentence.
15. Line 44 - The beginning of this sentence makes no sense, I think there is a typo.
16. Line 45 - “This” instead of ”the”.
17. Line 46 - Should be “several samples from diseased animals”
18. Line 48 - “were” instead of “that was”.
19. Line 49 - Add viruses after stomatitis.
20. Lines 55 and 56 - This sentence makes no sense.
21. Line 56 - Remove “The”
22. Line 60 - Should be current not currently and remove “the”.
23. Line 65 - Virus escaping is not the proper term.
24. Line 89 - This sentence is missing “long positive-sense single-stranded RNA” after nucleotides.
25. Line 104 - It should be live attenuated vaccine, otherwise it does not make sense to use the virus to eliminate SVA.
26. Line 110 - Pathway is not the proper term.
27. Line 111 - “Pathway of infection” is not the proper term.
28. Line 116 - This sentence makes no sense because is not constructed properly.
29. Line 120 to 121 - Same comment as the previous one.
30. Line 128 to 129 - The proper term is “days post-infection”
31. Line 142 - The sentence is not constructed properly: “To test whether recombinant VP0, VP1 and VP1 could assemble…”
32. Line 142 - Substitute “with” by “by”
33. Line 156 - “levels” not “level”
34. Line 172 - the proper sentence would be “there is no statistical difference between IgG1 and IgG2 levels….”
35. Line 198 to 199 - The syntax of this sentence is just wrong.
36. Line 199 - It is western blot not blotting.

Author Response

Response to Reviewer 1 Comments

Point 1: Please use a translation service as the quality of the English is very poor. All comments here are related to the highlighted text in the attached pdf file.

Response 1: Thanks for your comments. We modified scientific English regarding grammar, wording, interpunctuation in the article and thank International Science Editing (http://www.internationalscienceediting.com) for editing this manuscript.

Point 2: Lines 100 to 101 - It would be good to very briefly describe how the immunization was performed… intravenously, one dosis, adjuvants?? Just a brief description.

Response 2: We added the brief description in line 95: “SVA VLPs is emulsified by using adjuvant ISA 201 , inject  intramuscularly behind the root of the pig’s ear with one dose.”

Point 3: Line 108 - How can you talk about mimic’s natural infection using vaccination. This makes no sense, perhaps something got lost in the translation. This sentence makes no sense.

Response 3: We corrected the sentence in line 110: “By observing the incidence of pigs with three different routes of infection”

Point 4: Line 111 - This line makes no sense; you are talking about a vaccine and at the same time you are talking about infection. Do you mean symptoms? It is very confusing. Also, what are groups 1, 2, 3 and 4? These groups need to be defined.

Response 4: It was revised in Point 3. By observing the incidence of pigs with three different routes of infection, we studied the optimal route and dose of infection, not vaccine immunity. The definitions of Group 1, Group 2, Group 3, Group 4 and Group 5 are in Table 2 and materials and methods.

Point 5: Line 112 - The volume of the infection is not important, what it is important is the number of viral particles. You can inoculate with 1L of particles and if they are non-infections or if there is 1 particle per liter nothing will happen. Please clarify this. How did you determine that the infection rate is 100%? This applies to the rest of the paragraph. Talking about volumes is completely irrelevant, the important value is the number of plaques forming units, the multiplicity of infection or the number of genome copies per mL. Those experiments should be in terms of these quantities, otherwise it is impossible to replicate the experiments or to understand what is going on. Furthermore, this shows the the authors have very little understanding about basic virology, which is worrisome. A virologist will never make this mistake.

Response 5: Thanks for your suggestion. The dose of virus is indicated as PFU/mLin revised manuscript.

Point 6: Line 117 - Now the mention group 5 which has been not defined. Please see comment 3.

Response 6: The group 5 is a control group in which pigs were injected with PBS. We also described in section 4.5 of Materials and Methods in revised manuscript.

Point 7: Lines 124 to 125 - How would you observe clinical symptoms from anticoagulantes blood samples?

Response 7:  Sorry for making this mistake. It was revised in line 106-108 as “ the clinical scores of each pigs were recorded and the viral RNA was detected in the blood (from pigs #11, #12, #13, #14 and #15) with qRT–PCR.”

Point 8: Figure 1 - The resolution of this figure is poor, I cannot see the scale in figure 1D. Also, please draw a line indicating the limit of detection of you assay in figure 1D. That is, the copy number you get with the non-template control.

Response 8: Thanks for your suggestion. The Figure 1 was revised as followed.

Point 9: Figures - The resolution of all figures is really bad. 

Response 9: Thanks for your comments. The resolution of all pictures has been adjusted.

Point 10: Figure 2A and B - While cutting and pasting gels can be an acceptable procedure. What was done in figure 2A and 2B is unacceptable; you cannot copy a lane from a different experiment and then compare it with the ladder of other gel. I cannot accept these two gels; they need to be repeated so there is a ladder in the western blot. 

Response 10: Thanks for your good suggestion. In fact, we detected proteins in one Gel. After SDS-PAGE electrophoresis, half of gel was used for staining by Coomassie brilliant blue and another half gel was used for Western blot (WB) as shown as below original figure.

In previous version, we just combined the picture of SDS-PAGE with WB of specific proteins. According to your suggestion, we revised the Figure 2A as followed.

Point 11: Figure 2C - There axis are not labeled. Please label them.

Response 11: Sorry for making this error. The axis were labled as followed.

Point 12: Figure 2 - the letters in the figure legend dos not correspond to the images.

Response 12: Thank you for pointing out this error. We modified the figure legend as your suggestion.

Point 13: Line 145 to 146 - Do you mean a z-sizer? What is the polydispersity index of the measurements and standard deviation?

Response 13: Thanks for reviewer’ comments.

The nanoparticle size analyser assay means z-sizer, and the error revised in the new manuscript.

The polydispersity index (PDI) of the measurements were 0.194(pentamer) and 0.247(VLPs) and added as figure to supplementary information (C picture).

The standard deviation were 0.018 (pentamer) and 0.09 (VLPs). and added Error bar as figure to supplementary information (C picture).

Point 14: Line 147 - What is the standard deviation of the measurements?

Response 14: Thanks for reviewer’ comments. This question is answered in Point 13. And Transmission electron microscopy (TEM) not used the standard deviation of the measurements

Point 15: Section 2.3 - Do the particles contain RNA? 

Response 15: SVA VLPs in our research does not contain the viral RNA or other RNA. So we just detected the optical density of protein using UV-vis at 280 nm.

Point 16: Section 2.3 and 2.4 - In previous sections it was mentioned that there were five groups, suddenly this sections only mention three groups. What is going on? Please clarify.

Response 16: Thanks for reviewer’ comments. In Infectious routes and doses of SVA CH-HB-2017 strain experiments, pigs were divided into five groups as showed in table 1. In VLPs vaccine immunization experiment, three groups of pigs were used including inactivated vaccine group, the VLP vaccine group and control group.

Point 17: Lines 153 to 156 - What dosis of vaccine was used? How were the animals immunized (route)? 

Response 17: In section of Materials and Methods 4.5, it described as “inactivated vaccine group(n=5) was injected intramuscularly with 2 mL of inactivated SVA vaccine (10 ug). The VLP vaccine group(n=5) was injected intramuscularly with 2 mL of SVA VLP (50 μg) vaccine. Pigs in control group (n=5) were injected with PBS.

Point 18: Lines 160 to 161 - How much higher?

Response 18:  Sorry for unclear expression, It was revised in line 133-137. as “As shown in Fig.3A, the similar SVA-specific antibody levels were obtained in VLP group and the inactivated group. The antibody levels of the inactivated group slightly decreased at 49 days after immunization compared to VLP group (P < 0.01). the micro neutralization assay shown that higher neutralizing antibodies produced both by inactivated and VLP vaccine (Fig. 3B)”

Point 19: Lines 162 to 166 - Please give a brief description of the microneutralizaiton assay. More importantly, you needs to specify the dilution of the serums to know how powerful the neutralization activity of the vaccine is. This assay should mention the amount of virus used for the neutralization assay and all the tested dilutions; this is critical information, without it it might be that you need to have a 1:100 dilution of the serum to neutralize the virus for one group and a 1:1000 for the other group. Or it might be that only undiluted serum gives protection, all this information is key to understand to magnitude of the protective response.

Response 19: About microneutralization asssy of FMDV, there is criteria or formal protocal established by OIE ( World organisation for animal health).Hence, in our manuscript, we just described it in section 4.7 of Materials and Methods as “Serum samples were inactivated at 56°C for 30 min. Serum neutralization assays were then performed on IBRS-2 cells. Diluted serum samples were prepared starting with 1:4, repeating 4 times for each dilution. The diluted sera were incubated with 100TCID50 of SVA strain CH-HB-2017 for 1 h at 37°C. IBRS-2 cells (105 cells/mL, 50 μL) were then added to  the wells of 96-well plates, cultured in an incubator at 37°C under 5% CO2 for 72 h, and  observed for cytopathic effects (CPEs) under an inverted microscope. The neutralizing antibody titer was calculated by the Reed-Müench method. In brief, 100% CPE of cells in the tested serum wells is judged as negative, and more than 50% of cells are protected as positive. Finally, calculate the serum dilution that can protect 50% of the cell wells from cytopathic, and this dilution is the serum neutralizing antibody titer”.

Point 20: Lines 172 to 174 - by how much?

Response 20: Thanks for your suggestion. It was revised in line 132-148 as “Serum specific antibody levels were detected by indirected- ELISAs at 14, 21, 35 and 49 days after immunization. As shown in Fig.3A, the similar SVA-specific antibody levels were obtained in VLP group and the inactivated group. The antibody levels of the inactivated group slightly decreased at 49 days after immunization compared to VLP group (P < 0.01). The micro neutralization assay shown that higher neutralizing antibodies produced both by inactivated and VLP vaccine (Fig. 3B).

The degrees of cellular immune responses (Th1) and humoral immune responses (Th2) are represented by the levels of IgG2a and IgG1 antibodies, respectively. Serum IgG1 and IgG2a antibodies levels were detected by ELISAs at 14, 21, 35 and 49 days after immunization (Fig.3C, 3D). The IgG1 antibody levels obtained by the inactivated vaccine was significantly higher than that of the VLPs vaccine at 14(p<0.01), 21(p<0.01), 35(p<0.05) and 49(p<0.01) days after immunization. However, The VLP vaccine IgG2a antibody level were similar and no statistical difference to inactivated vaccine at 14 and 21 days after immunization (p=ns). But, the antibody levels of inactivated vaccine slightly increased at 35 (p<0.05) and 49 days(P<0.0001) after immunization compared to VLP vaccine group. This result indicated that strong humoral immune responses were induced by the VLP vaccine and inactivated vaccine, and was more obvious in the inactivated vaccine.”

 Point 21: Lines 183 to 184 - by how much?

Response 21: It was revised in line 150-154 as “To further evaluate the VLP vaccine-induced cellular immune response, the INF-γ levels were detected. As shown in Fig. 4, INF-γ produced by the inactivated vaccine and VLP vaccine has no significant difference at 14, 21,35 days after immunization (p=ns). However, the duration of the inactivated vaccine inducing cellular immunity is significantly longer than 155 that of the VLP vaccine (p<0.01)”.

Point 22: Lines 187 to 188 - Before the groups were, vaguely, identified by numbers now is by letters. This is totally confusing.

Response 22: Sorry for making confusion. It was revised in line 156-162 as “To determine the protective effect of the VLP vaccine in pigs, all pigs were challenged  with SVA CH-HB-2017 strain infection via intramuscular (3 mL, 7x107.8 PFU/mL) and  intranasal (1.5 mL into each nostril, 7x107.8 PFU/mL) routes at 49 days after immunization. Clinical signal was recorded and scored according to Table 1. As shown in Table 3, the five pigs in the PBS group appeared clinical symptoms within 3 days after challenge. Lesions were observed on foot hooves of all pigs. Pigs vaccinated with VLP and the inactivated vaccine has no clinical signs and no RNA in blood samples was detected”.

Point 23: Important observation: from the purified sample what is the proportion of assembled VLPs and partially assembled VLPs. This information is key. Also please, explain how you determined this.

Response 23: It is a good question. In our research, the proportion of assembled VLPs and partially assembled VLPs were detected by sucrose density gradient centrifugation and confirmed the antigen content in the fractions of curve peak by the BCA protein assay kit. Then it was calculated by ratio of assembled VLPs to partially assembled VLPs.

Point 24: Besides all the problems mentioned in the previous line and in the following section the authors do not clarify the composition of the vaccine (i.e., VLP concentration) and purity (fraction of proteins as VLPs and as incomplete VLPs in the vaccine). This is a crucial point for the formulation of the vaccine. At least they should mention the total amount of protein in each immunization and how this compares with the inactivated vaccine. Did they use the same amount of protein in the VLPs and inactivated vaccine? If not, why?

Response 24: Thanks for your question.

On one hand, in our study, the assembly ratio of VLPs usually is stable, which is confirmed by sucrose density gradient centrifugation together with the BCA protein assay. So, in animal experiment, we just indicated the total quantity of purified VLPs for vaccination, which is described in section 4.2 and 4.3.

On the other hand, it already confirmed, if the assembly ratio of VLPs is stable, certain quantity of purified VLPs protein must provide the protection to animals in our previous studies and other papers (Gabriela M. Escalante etal, doi:10.3390/vaccines8020169. Guo et al. Veterinary Research 2013, 44:48). Hence, in this study, it was described as quantity of purified VLPs, namely, one pig was immunized with 50ug of purified VLPs.

For inactivated SVV, there is a general approach to detect the quantity of 146s protein which can be indicated as purified whole virus. Hence, in our study, we immunized each pig with 10 ug purified SVV.

VLPs vaccines and inactivated vaccines use immunization doses with different antigen content. VLPs vaccine content is determined using our previous study, 50 ug /pig. The recommended dose of the commercial FMDV inactivated vaccine is not less than 3 ug, so we chose a dose of 10 ug to ensure that the pigs can be completely protected from SVA infection.

Point 25: The lack of experimental information makes impossible to future replicate these results in other labs. Please be careful detailing all experimental information.

Response 25: Thanks for your suggestion. We described the experiment in section of Materials and Methods in more details so that other lab can repeat this experiment.

Point 26: Lines 231 to 232 - I don’t understand how they can conclude this. These are different viruses.

Response 26: Sorry for making this error. Although FMDV and SVA are different virus, both them belong to the Picornaviridae family. In reference 29, it showed that SVA predominant antigen epitope exists in VP2 protein. However, predominant antigen epitope of FMDV exist in VP1 protein, which is confirmed by many researches.  Hence, in our study, combined with animal immunization and antibody level, it is speculated that the high antibody level induced by SVA may be related with different antigenic sites between SVA and FMDV.

Point 27: Lines 235 to 236 - this was not clear int he previous sections. Please explicitly say by how much.

Response 27: Thanks for your remind. We revised this setence in the line 213-218 as “Th1-type cellular immune response caused by IFN-γ is mainly mediated by T lymphocytes, which primarily destroy endogenous antigens and are therefore essential for viral clearance.  In the present study, the VLPs induced similar levels of INF-γ than the inactivated vaccine(p=ns) before 35 days after immunization. This virus mainly infects the host by inhibiting the production of INF-γ to escape the immune response [10]”.

Point 28: Line 237 - Virus infect tissues, they do not parasite. 

Response 28: Thanks for point out this error. It was revised in line 218-219 as “It mainly exists the tonsils of pigs after infection [33], leading to continuous infection of the host

Point 29: Lines 240 to 241 - What does full dose mean? What is the composition of a dose?

Response 29: It was revised in line 220-224 as “ In this study, the inactivated vaccine(10 ug)and the VLP vaccine (50 ug)can full  protected pigs from SVA infection. However, a more suitable immunization dose was not determined, future studies should focus on the development of excellent adjuvants and suitable immunization dose, as well as the optimization of the infectious dose”.

Point 30: Lines 250 to 251 - this was not mentioned in the results and should be.

Response 30: Thanks for reviewer’ comments. According to previous studies, we tried 5 pigs, but did not produce clinical symptoms through this route. So we only explained in the discussion. ( not published)

Point 31: Line 254 - The dose of the virus should be expressed in PFU/mL, please convert the TCID 50/ml to PFU/mL.

Response 31: thanks for reviewer’ comments. This Thank you for your advise. The virus titer question is was revised in the article.

Point 32: Line 259 to 260 - This statement makes no sense. In the previous sentences they explain that they cannot achieve 100% infection, thus it makes no sense to suggest further dilutions. What you need are more concentrated viral samples. 

Response 32: Thanks for reviewer’ comments. This question is revised in the article. It was revised in line 240-242 as “Our results suggested that it may be future direction to further concentrate viral sample in order to achieve SVA infection in pigs and developed typical symptom via marginal ear vein”

Point 33: A big pitfall in this article is the lack of information about the ideal vaccine dilution that can protect the pigs. 

Response 33: Thanks for reviewer’ comments. Thanks for your suggestion. It is the first report about SVA VLPs vaccine. Hence, in this study, we just focused on whether the SVA VLPs can induce effective immune response and proved the protection for pigs. What optimal dose of SVA VLPs is or how much dose of VLPs at least can provide 100% protection for pigs will be our next study, which will be published as commercial vaccine.

Point 34: Lines 297 - If the virus was inactivated then how can you make three passages? This data is key, please put in a figure.

Response 34: It was revised in line 274-279 as “The SVA inactivated vaccine was prepared according to previous studies [36, 37]. In brief, the virus was collected and extracted, after IBRS-2 cells were infected with SVA for 10 h (until 100% cytopathic effect). The viral suspension was inactivated by binary ethylenimine at 30°C for 28 h. The inactivation effect of the virus is verified by reverse transcription (RT)- quantitative PCR (qPCR). The antigen was further purified by ultracentrifugation, and majored antigen peak was quantitated by the BCA protein assay kit”.

Point 35: Lines 300 to 303 - Please see comments 17, 24 and 29.

Response 35: It was revised in line 280-283 as “VLPs and inactivated antigen were emulsificated with equal volume of ISA 201 adjuvant (Montanide). ISA 201 was used as the oil phase. The oil phase and inactivated SVA fluid or VLPs were homogenized with a T 10 basic Ultra-Turrax® homogenizer (IKA, Beijing China) at 8,000 rounds per minute for 30 min”.

Point 36: Section 4.5 - The groups are not explained in the main text. please do it.

Response 36: Thanks for your advice. The groups have been explained in the main text manuscript and table 2.

Point 37: Lines 316 to 321 - This is fundamental error in the experimental design. They should have used innocuous with different amount of SVA so that each infection protocol delivered the same amount virus. This explain why they were not Abe to have 100% infection using different routes of infection.

Response 37: It was revised and explained in line 225-242 as “The pathogenic mechanism of SVA has been reported [34], but the dose and pathway of viral infection have not yet been established. However, the development of vaccines requires both a suitable challenge dose and route. Different strains may also favor different paths of infection. For example, oronasal infection has been commonly used in previous studies. A dose of 10 mL at a 107.79TCID50/mL can result in successful SVA infection, although this is a very high dose [34]. However, in this study, infection via oronasal was failed at this dose (Date no shown), which may be attributable to differences in the procedures and strains used. The failure of animal experiments is very costly. Therefore, we changed the infection strategy and used nasal (1.5 mL into each nostril, 7x107.8 PFU/mL) and intramuscular (3 mL, 7x107.8 PFU/mL) injections to achieve 100% infection. An intramuscular injection of 6 mL (7x107.8 PFU/mL) alone was not ideal because it only reached 80% infection. There is no report of disease with SVA via the intramuscular routes, which may be attributable to its slow absorption and the complex bio environment. SVA infection via the marginal ear vein is theoretically the fastest way to infect pigs, but the complexity of the operation makes it very challenging when more than 3 mL (7x107.8 PFU/mL) dose is injected, although it surprisingly achieved an infection rate of 80%. Our results suggested that further concentrated viral samples may be a future direction in which to explore the optimal dose and routes for SVA infection”.

Point 38: Lines 328 to 331 - Is the does of inactivated SVA vaccine the same as the VLPs? In other words, are 10^-8.8 the same as 50 ug? If not then this a problem. Please explain how you determined this.

Response 38: 10^-8.8 is the titer of the virus, which has nothing to do with the amount of antigen. The 10 ug dose of the inactivated SVA vaccine is obtained by using standard methods to detect 146s, which is a method that refers to the international standard for FMDV vaccine.

Point 39: Lines 349 to 357 - Please describe the methods.

Response 39: We describe the methods in detail in section 4.7, 4.8, 4.9.

Point 40: Section 4.9 - Please indicate the limit of detection of this assay.

Response 40: This limit of detection of this assay is added in the Section 4.9 and figure 1.

Response to Editorial comments

Point 1: Line 18 - see comment

Response 1: Thanks for editorial comments. It was revised.

Point 2: Lines 22 to 25 - This sentence is hard to understand. The syntaxes should be improved.

Response 2: The sentence was revised as “Senecavirus A (SVA) is the pathogen that has recently caused porcine idiopathic vesicular disease (PIVD). The clinical symptoms of PIVD are similar to those of acute foot-and-mouth disease and also can result in the death of newborn piglets, thus entailing economic losses.”. in Line 21-24 of new version.

Point 3: Line 26 - remove the comma

Response 3: It was removed in in Line 21-24 of new version.

Point 4: Line 27 - were instead of was (VLPs is plural)

Response 4: It was instead of “was” in Line 26 of new version.

Point 5: Line 28 - What technique do you mean by “nanoparticle detection”?

Response 5: Thanks for editorial comments, the “nanoparticle detection” has revised to “zetasizer nano” in Line 27 of new version.

Point 6: Lines 29 to 30 - I don’t think that parameters of infection pathway is the proper term. Please use a better term to describe what you did. This term is so confusing that it is not clear what do you mean.

Response 6: The sentence was revised as “Meanwhile, SVA CH-HB-2017 strain was used to infect pigs and to determine of infection routes and content.” in Line 27-28 of new version.

Point 7: Line 30 - The proper term is immunized not injected. 

Response 7: It was removed in Line 29 of new version.

Point 8: Line 31 - Is oil the vaccine carrier or the adjuvant? I would think is the vaccine carrier.

Response 8: Thanks for editorial comments and this question was revised as suggested. The sentence was revised as “ISA 201 adjuvant” in Line 30 of new version.

Point 9: Line 31 to 32 - Remove the hyphen between vaccine and induced, otherwise the rest of the sentence makes no sense. Perhaps use vaccination or immunization rather than vaccine. 

Response 9: We thank for editorial comments and the sentence was revised to clear puzzling in Line 31-32 of new version.

Point 10: Line 35 to 36 - The statement about IgG1 and IgG2 does not belong to an abstract. The important information is contained before the comma.

Response 10: The sentence was removed in Line 33 of new version.

Point 11: Line 37 - I am not sure that protection rate is the proper term. A rate is the change of parameter with respect of another parameter. Perhaps, levels of protection would be a better term.

Response 11: The sentence was revised as “levels of protection” in Line 34 of new version.

Point 12: Line 38 - “Good” immunogenicity is not the right term. What about “is highly immunogenic”?

Response 12: Thanks for editorial comments. The “Good” has revised to “is highly immunogenic” in Line 37 of new version.

Point 13: Line 39 - This sentence is incomplete. “Of SVA, compared to the inactivated vaccine”.

Response 13: The sentence as “Of SVA, compared to the inactivated vaccine” has added in Line 37 of new version.

Point 14: Line 43 - There should be a reference at the end of the first sentence.

Response 14: The reference has added in Line 41 of new version.

Point 15: Line 44 - The beginning of this sentence makes no sense, I think there is a typo. 

Response 15: This sentence was detected as suggested in Line 41 of new version.

Point 16: Line 45 - “This” instead of ”the”.

Response 16: The “the” has revised to “this” in Line 41 of new version.

Point 17: Line 46 - Should be “several samples from diseased animals”.

Response 17: Thanks for editorial comments. We revised the sentence “several diseased samples” to “several samples from diseased animals” in Line 43 of new version.

Point 18: Line 48 - “were” instead of “that was”.

Response 18: The “was” has revised to “were” in Line 44 of new version.

Point 19: Line 49 - Add viruses after stomatitis.

Response 19: We add viruses after stomatitis in Line 46 of new version.

Point 20: Lines 55 and 56 - This sentence makes no sense.

Response 20: This sentence was removed in Line 51-52 of new version.

Point 21: Line 56 - Remove “The”

Response 21: This sentence was removed in Line 52 of new version.

Point 22: Line 60 - Should be current not currently and remove “the”.

Response 22: This sentence was revised in Line 56 of new version.

Point 23: Line 65 - Virus escaping is not the proper term. 

Response 23: The sentence was removed as “Virus escaping” and revised the sentence in Line 61 of new version.

Point 24: Line 89 - This sentence is missing “long positive-sense single-stranded RNA” after nucleotides.

Response 24: The sentence was added as “long positive-sense single-stranded RNA” after nucleotides.” in Line 56 of new version.

Point 25: Line 104 - It should be live attenuated vaccine, otherwise it does not make sense to use the virus to eliminate SVA.

Response 25: The sentence was added as “the attenuated or inactivated vaccine” in Line 99-100 of new version.

Point 26: Line 110 - Pathway is not the proper term.

Response 26: The “Pathway” has revised to “routes” in Line 103 of new version.

Point 27: Line 111 - “Pathway of infection” is not the proper term.

Response 27: The “Pathway” has revised to “routes” in the article. And we have completely rewritten this part of the conclusion to express the conclusion in more concise sentences as“By observing the incidence of pigs with three different routes of infection, as shown in Table 2, it is found that all pigs infected SVA via intramuscular (3 mL, 7x107.8 PFU/mL) and intranasal (1.5 mL into each nostril, 7x107.8 PFU/mL) presented clinical signs (Fig. 1A). The clinical scores of each pigs were recorded and the viral RNA was detected in the blood (from pigs #11, #12, #13, #14 and #15) with qRT–PCR. As shown in Fig. 1A, the highest level of the virus RNA was detected at 3 dpi, Afterwards, it was gradually declined and undetectable at 7 dpi. Vesicles appear on the nose or hoof at 4 to 6 dpi, then ulcerated at 7 dpi (Fig. 1B).The mean rectal temperature was higher than that of the control group (Fig. 1C).” in Line 106-113 of new version.

Point 28: Line 116 - This sentence makes no sense because is not constructed properly. 

Response 28: Thanks for editorial comments and this question was revised as suggested. we have completely rewritten this part of the conclusion to express the conclusion in more concise sentences as“By observing the incidence of pigs with three different routes of infection, as shown in Table 2, it is found that all pigs infected SVA via intramuscular (3 mL, 7x107.8 PFU/mL) and intranasal (1.5 mL into each nostril, 7x107.8 PFU/mL) presented clinical signs (Fig. 1A). The clinical scores of each pigs were recorded and the viral RNA was detected in the blood (from pigs #11, #12, #13, #14 and #15) with qRT–PCR. As shown in Fig. 1A, the highest level of the virus RNA was detected at 3 dpi, Afterwards, it was gradually declined and undetectable at 7 dpi. Vesicles appear on the nose or hoof at 4 to 6 dpi, then ulcerated at 7 dpi (Fig. 1B).The mean rectal temperature was higher than that of the control group (Fig. 1C).” in Line 106-113 of new version.

Point 29: Line 120 to 121 - Same comment as the previous one.

Response 29: Thanks for editorial comments and this question was revised as suggested. We have completely rewritten this part of the conclusion to express the conclusion in more concise sentences as “By observing the incidence of pigs with three different routes of infection, as shown in Table 2, it is found that all pigs infected SVA via intramuscular (3 mL, 7x107.8 PFU/mL) and intranasal (1.5 mL into each nostril, 7x107.8 PFU/mL) presented clinical signs (Fig. 1A). The clinical scores of each pigs were recorded and the viral RNA was detected in the blood (from pigs #11, #12, #13, #14 and #15) with qRT–PCR. As shown in Fig. 1A, the highest level of the virus RNA was detected at 3 dpi, Afterwards, it was gradually declined and undetectable at 7 dpi. Vesicles appear on the nose or hoof at 4 to 6 dpi, then ulcerated at 7 dpi (Fig. 1B).The mean rectal temperature was higher than that of the control group (Fig. 1C).” in Line 106-113 of new version.

Point 30: Line 128 to 129 - The proper term is “days post-infection”

Response 30: We have completely rewritten this part of the conclusion to express the conclusion in more concise sentences as “By observing the incidence of pigs with three different routes of infection, as shown in Table 2, it is found that all pigs infected SVA via intramuscular (3 mL, 7x107.8 PFU/mL) and intranasal (1.5 mL into each nostril, 7x107.8 PFU/mL) presented clinical signs (Fig. 1A). The clinical scores of each pigs were recorded and the viral RNA was detected in the blood (from pigs #11, #12, #13, #14 and #15) with qRT–PCR. As shown in Fig. 1A, the highest level of the virus RNA was detected at 3 dpi, Afterwards, it was gradually declined and undetectable at 7 dpi. Vesicles appear on the nose or hoof at 4 to 6 dpi, then ulcerated at 7 dpi (Fig. 1B).The mean rectal temperature was higher than that of the control group (Fig. 1C).” in Line 106-113 of new version.

Point 31: Line 142 - The sentence is not constructed properly: “To test whether recombinant VP0, VP1 and VP1 could assemble…”

Response 31: The sentence is revised as constructed “To test whether recombinant VP0, VP1 and VP1 could assemble…”in Line 124 of new version.

Point 32: Line 142 - Substitute “with” by “by” 

Response 32: This sentence was revised in Line 125 of new version.

Point 33: Line 156 - “levels” not “level” 

Response 33: This sentence was revised in Line 134 of new version.

Point 34: Line 172 - the proper sentence would be “there is no statistical difference between IgG1 and IgG2 levels….”

Response 34: The sentence was added as “there is no statistical difference between IgG1 and IgG2 levels…” in Line 146-147 of new version.

Point 35: Line 198 to 199 - The syntax of this sentence is just wrong.

Response 35: the sentence was revised in Line 184-185 of new version.

Point 36: Line 199 - It is western blot not blotting.

Response 36: the sentence was revised.

Reviewer 2 Report

The authors developed a vaccine for Senecavirus A (SVA) based on virus-like particles and tested it on pigs who were then challenged with SVA. The vaccine provided 100% protection and created a similar immune response as the inactivated virus vaccine. SVA vaccines based on VLPs may be preferable to inactivated SVA vaccines due to reduced adverse reactios to vaccination and decreased potential to spread SVA virus through incompletely inactivated virus. The study was well designed, executed, and the manuscript written clearly. I have no concerns.

Line 108: I believe the term should be inoculation sites instead of vaccination sites. Line 111: While the groups are defined later in the methods, it would be helpful to the reader to briefly describe them here. Also I would suggest changing pathway of infection to inoculation route or sites. Lines 112-114: How many viral particles or other appropriate measure /mL were in the 3mL and 6mL doses? Density is more important than volume. Lines 124-125: It is unclear what the authors are trying to say here beyond collecting blood samples. Are the authors trying to correlate symptoms with level of viral RNA as is reflected in Figure 1D. Lines 158-161. This is misleading as written. The VLP group was only significantly higher than the inactivated vaccine group at 49 day. Please clarify. Lines 327-331: How many pigs are in each group? Line 359: Given the small sample size a nonparametric statistical method should be used (Kruskal Wallis/Mann-Whitney) The labels A-E for Figure 2 do not correspond to the images and there is not an explanation for C. C is also lacking axis titles. Figures 3 and 4 should include the number of pigs included in each group.

Author Response

Response to Reviewer 2 Comments

The authors developed a vaccine for Senecavirus A (SVA) based on virus-like particles and tested it on pigs who were then challenged with SVA. The vaccine provided 100% protection and created a similar immune response as the inactivated virus vaccine. SVA vaccines based on VLPs may be preferable to inactivated SVA vaccines due to reduced adverse reactios to vaccination and decreased potential to spread SVA virus through incompletely inactivated virus. The study was well designed, executed, and the manuscript written clearly. I have no concerns.

Point 1: Line 108: I believe the term should be inoculation sites instead of vaccination sites.

Response 1: Thanks for your suggestion. It was revised as “By observing the incidence of pigs with three different routes of infection, as shown in  Table 2, it is found that all pigs infected SVA via intramuscular (3 mL, 7x107.8 PFU/mL) and  intranasal (1.5 mL into each nostril, 7x107.8 PFU/mL) presented clinical signs (Fig. 1A). The clinical scores of each pigs were recorded and the viral RNA was detected in the blood (from pigs #11, #12, #13, #14 and #15) with qRT–PCR”

Point 2: Line 111: While the groups are defined later in the methods, it would be helpful to the reader to briefly describe them here. Also, I would suggest changing pathway of infection to inoculation route or sites.

Response 2: It was revised as “By observing the incidence of pigs with three different routes of infection, as shown in  Table 2, it is found that all pigs infected SVA via intramuscular (3 mL, 7x107.8 PFU/mL) and  intranasal (1.5 mL into each nostril, 7x107.8 PFU/mL) presented clinical signs (Fig. 1A). The  clinical scores of each pigs were recorded and the viral RNA was detected in the blood (from pigs #11, #12, #13, #14 and #15) with qRT–PCR”

Point 3: Lines 112-114: How many viral particles or other appropriate measure /mL were in the 3mL and 6mL doses? Density is more important than volume.

Response 3:  It was revised as “it is found that all pigs infected SVA via intramuscular (3 mL, 7x107.8 PFU/mL) and intranasal (1.5 mL into each nostril, 7x107.8 PFU/mL) presented clinical signs (Fig. 1A)”

Point 4: Lines 124-125: It is unclear what the authors are trying to say here beyond collecting blood samples. Are the authors trying to correlate symptoms with level of viral RNA as is reflected in Figure 1D.

Response 4: It was revised as “As shown in Fig. 1A, the highest level of 110 the virus RNA was detected at 3 dpi, Afterwards, it was gradually declined and undetectable 111 at 7 dpi. Vesicles appear on the nose or hoof at 4 to 6 dpi, then ulcerated at 7 dpi (Fig. 1B).The 112 mean rectal temperature was higher than that of the control group (Fig. 1C)”.

Point 5: Lines 158-161. This is misleading as written. The VLP group was only significantly higher than the inactivated vaccine group at 49 day. Please clarify.

Response 5: It was revised as “The degrees of cellular immune responses (Th1) and humoral immune responses (Th2) are represented by the levels of IgG2a and IgG1 antibodies, respectively. Serum IgG1 and  IgG2a antibodies levels were detected by ELISAs at 14, 21, 35 and 49 days after  immunization (Fig.3C, 3D). The IgG1 antibody levels obtained by the inactivated vaccine  was significantly higher than that of the VLPs vaccine at 14(p<0.01), 21(p<0.01), 35(p<0.05) and 49(p<0.01) days after immunization. However, The VLP vaccine IgG2a antibody level  were similar and no statistical difference to inactivated vaccine at 14 and 21 days after  immunization (p=ns). But, the antibody levels of inactivated vaccine slightly increased at 35 (p<0.05) and 49 days(P<0.0001) after immunization compared to VLP vaccine group. This result indicated that strong humoral immune responses were induced by the VLP vaccine and inactivated vaccine, and was more obvious in the inactivated vaccine”.

Point 6: Lines 327-331: How many pigs are in each group?

Response 6: It was revised as “To evaluate the efficiency of the SVA VLP vaccine, 9-week-old finishing pigs (approximately 35 kg each) were purchased from conventional breeding/fattening farms and kept in three rooms, with access to food and water ad libitum. They were all negative for 312 FMDV and SVA antibodies at the start of the experiment. The pigs (total 15 pigs) were randomly assigned to three groups: inactivated vaccine group(n=5) was injected intramuscularly with 2 mL of inactivated SVA vaccine (10 ug). The VLP vaccine group(n=5) was injected intramuscularly with 2 mL of SVA VLP (50 μg) vaccine. Pigs in control group (n=5) were injected with PBS. All pigs were boost vaccinated at 21 days. Animal serum samples were collected at 0, 14, 21, 35, and 49 days and stored at -80°C. At 49 days, all pigs were challenged with SVA CH-HB-2017 strain via the intramuscular (3 mL, 7x107.8 PFU/mL) and intranasal routes (1.5 mL into each nostril, 7x107.8 PFU/mL)”.

Point 7: Line 359: Given the small sample size a nonparametric statistical method should be used (Kruskal Wallis/Mann-Whitney) The labels A-E for Figure 2 do not correspond to the images and there is not an explanation for C. C is also lacking axis titles. Figures 3 and 4 should include the number of pigs included in each group.

Response 7: Thanks for reviewer’ comments, the question was revised as suggested.

Because more than two groups are compared in figure 3 and figure 4. So we use the 2-way ANOVA analysis.

The figure 2 has revised in the article.

Figuer 3 and 4 the number of pigs were added as “Levels of serum antibody were detected from pigs vaccinated with SVA VLPs (n=5), inactivated vaccines (n=5) and PBS (n=5) by 0 days, 14 days, 21 days, 35 days and 49 days post-immunization. ”and“Serum were collected from pigs vaccinated with SVA VLPs (n=5), inactivated vaccines(n=5) and PBS(n=5)at 0 days, 14 days, 21 days, 35 days and 49 days post-immunization.”in line 512-516 and line 538-539 of new version.

Reviewer 3 Report

Dear Authors,

the manuscript submitted treats a very interesting and current topic. 

Best regards

Author Response

Response to Reviewer 3 Comments

Point 1: The manuscript submitted treats a very interesting and current topic.

Respond: Thanks for reviewer’ comments.

Reviewer 4 Report

This article is very poorly written, much too long, often repetitive and even at times redundant! Its style is too often pigeon english!

The figures are hard to read and understand. Some, like the electron mcroscopy figures (2D and 2E) are just useless!

The tables are uselessly repetitive and of pretty limited interest!!

How come the titers of the quoted references are not given in the "References" list?

Altogether, the paper needs thorough rewriting

Author Response

Response to Reviewer 4 Comments

Point 1: This article is very poorly written, much too long, often repetitive and even at times redundant! Its style is too often pigeon english!

Response 1: We appreciate the reviewer’s comments. We modified scientific English regarding grammar, wording, interpunctuation in the article and thank International Science Editing (http://www.internationalscienceediting.com) for editing this manuscript.

Point 2: The figures are hard to read and understand. Some, like the electron mcroscopy figures (2D and 2E) are just useless!

Response 2: The figures were revised the layout and clarity of the picture in order to better understand the content of the picture.

Point 3: The tables are uselessly repetitive and of pretty limited interest!!

Response 1: Thanks for reviewer’ comments, Table 1 in this study is to summarize the scores of clinical symptoms, Table 2 is to better explain which was optimal route of infection and dose, and Table 3 is to summarize the protection level of pigs after vaccination. These tables still have a certain value.

Point 4: How come the titers of the quoted references are not given in the "References" list?

Response 1: The References were revised the format of references in the text.

Point 5: Altogether, the paper needs thorough rewriting

Response 1: Thanks for reviewer’ comments. We rewritten the abstract, conclusion, discussion and conclusion of the article.

Round 2

Reviewer 1 Report

appreciate all the responses from he authors. Still I have some questions:

  1. Point 23. They explain how they can calculate the proportion of assembled and partially assembled VLPs. However, they do not answer the question; what is the proportion of assembled to partially assembled VLPS?. This is a key point as variations in this ratio might affect the reproducibility of the vaccine.
  2. Point 24. This is related to the previous point. Sayin that the ratio is stable is not good enough. Please, specify this ratio. In fact, they can do this very easily; then can take th data from figure 2b, fit the peaks for the pentamer and VLPs as two gaussians, calculate the area of each gaussian, added them up, and take the area of the VLPs over the sum.
  3. Response 34. Again, the refer to the VLPs or inactivated virus as “antigen”, This is not an acceptable term; an antigen is a substance which induces an immune response. However, viruses and VLPs are not a single antigen, they have multiple antigenic regions. In other words, a single VLPs can present different antigens to the immune system, as the response to each of these antigens can be different. Hence, please do not refer to the VLPs or inactivated viruses are antigens. This term is just wrong. Please replace this term across the whole article.

Comments:

  1. Line 229 . Please convert the TCID50/mL to PFU/mL, this will give the article a sense of consistency acorns the different quantities been used (like those in lines 233 and 234).
  2. Please read point 3. VLPs and viruses should not be called antigens; there are multiple antigenic regions in each of these particles.
  3. Figure 1; again the quality of the image is very poor; it is imposible to read the axis. Please fix it, it has to be crispy clear. Also, in 1A and 1B use an farro to point at at what you want us to pay attention to.
